Psychrophrynella glauca sp. n., a new species of terrestrial-breeding frogs (Amphibia, Anura, Strabomantidae) from the montane forests of the Amazonian Andes of Puno, Peru

Catenazzi Alessandro 1 2 acatenazzi@gmail.com
Ttito Alex 3 4
1 Department of Biological Sciences, Florida International University , Miami, FL , USA
2 Centro de Ornitología y Biodiversidad , Lima , Perú
3 Museo de Biodiversidad del Perú , Cusco , Perú
4 Museo de Historia Natural, Universidad Nacional de San Antonio Abad , Cusco , Perú
Parra Olea Gabriela
Electronic publication date: 2018 Feb 27
Publication date: 2018
Volume: 6
Electronic Location ID: e4444
Received 2017 Dec 1; Accepted 2018 Feb 12
Copyright: © 2018 Catenazzi and Ttito
Copyright year: 2018
Copyright holder: Catenazzi and Ttito
License: This is an open access article distributed under the terms of the Creative Commons Attribution License, which permits unrestricted use, distribution, reproduction and adaptation in any medium and for any purpose provided that it is properly attributed. For attribution, the original author(s), title, publication source (PeerJ) and either DOI or URL of the article must be cited.
License URL: https://creativecommons.org/licenses/by/4.0/

Keywords: Cloud forest, Frog, Bioacoustics, Carabaya, Ollachea, Leaf litter amphibian, 16S rRNA, Taxonomy, Holoadeninae, Terrarana

Funding: The Eppley Foundation Wildlife Acoustics Chicago Board of Trade Endangered Species Fund This work was supported by grants from The Eppley Foundation, Wildlife Acoustics, and the Chicago Board of Trade Endangered Species Fund. The funders had no role in study design, data collection and analysis, decision to publish, or preparation of the manuscript.

==============================
We describe a new species of small strabomantid frog (genus Psychrophrynella) from a humid montane forest in the Peruvian Department of Puno. Specimens were collected at 2,225 m a.s.l. in the leaf litter of primary montane forest near Thiuni, along the Macusani–San Gabán road, in the province of Carabaya. The new species is assigned to Psychrophrynella on the basis of morphological similarity, including presence of a tubercle on the inner edge of the tarsus, and call composed of multiple notes. We also include genetic distances for 16S rRNA partial sequences between the new species and other strabomantid frogs. The species with lowest genetic distances are Psychrophrynella chirihampatu and Psychrophrynella usurpator. Psychrophrynella glauca sp. n. is readily distinguished from the three other species of Psychrophrynella (Psychrophrynella bagrecito, P. chirihampatu, and P. usurpator) by its small size, and by having belly and ventral surfaces of legs reddish-brown or red, and chest and throat brown to dark brown with a profusion of bluish-gray flecks. The new species is only known from its type locality. With the discovery of P. glauca, the geographic distribution of Psychrophrynella is extended to the Department of Puno, where it was no longer represented after the description of the genus Microkayla. Furthermore, the Cordillera de Carabaya is the only mountain range known to be home to four of the seven genera of Holoadeninae (Bryophryne, Microkayla, Noblella, and Psychrophrynella), suggesting an intriguing evolutionary history for this group in southern Peru.

Introduction

Frogs in the genus Psychrophrynella are small, terrestrial-breeding terraranas that had originally been placed in the genus Phrynopus (Lynch, 1986). These high-Andean terraranas are very difficult to characterize morphologically, and molecular analyses later revealed that these species are closely related to Barycholos, Bryophryne, and Holoaden of the subfamily Holoadeninae, and not to Phrynopus and related forms within Strabomantinae (Hedges, Duellman & Heinicke, 2008). Thus, these forms were assigned to the new genus Psychrophrynella, with Psychrophrynella bagrecito as the type species (Hedges, Duellman & Heinicke, 2008). Until recently, the genus contained 23 species, four species from the Peruvian Departments of Cusco and Puno (Catenazzi & Ttito, 2016), and 19 from Bolivia. Following the description of the genus Microkayla, which contains all Bolivian species formerly assigned to Psychrophrynella (and the southern Peruvian species Microkayla boettgeri, Microkayla chapi, and Microkayla chilina), the genus Psychrophrynella presently contains only three species: P. bagrecito, Psychrophrynella chirihampatu, and Psychrophrynella usurpator (De la Riva et al., 2017).

The three species of Psychrophrynella are Peruvian endemics restricted to the Amazonian slopes of the Andes in the upper Kosñipata, Quespillomayo, Japumayo, and Marcapata valleys in the Department of Cusco, where they inhabit humid grasslands and montane forests from 1,770 to 3,600 m a.s.l. (Catenazzi & Ttito, 2016; Duellman & Lehr, 2009; von May et al., 2017). These small frogs inhabit the leaf litter and the layer of terrestrial mosses, and thus require considerable effort to be detected, for example through intensive search within leaf litter plots (Catenazzi et al., 2011). Most of the eastern valleys of the Andes in the southern Peruvian Departments of Cusco and Puno have been poorly explored, with few locations surveyed by using leaf litter plots, and are likely to contain many unnamed species of Psychrophrynella and other Holoadeninae (Catenazzi & von May, 2014).

The taxonomy of Holadeninae has undergone frequent changes over the past decade (reviewed in De la Riva et al. (2017)), in part because of our limited understanding of the phylogenetic relationships among its members. As new species are discovered, and our understanding is accrued, it is likely that phylogenetic relationships will be revised again. Furthermore, the type species of Psychrophrynella, P. bagrecito shares several morphological traits with the type species of Noblella, Noblella peruviana (De la Riva, Chaparro & Padial, 2008; Lehr, 2006), suggesting that the two species might be closely related. However, genetic sequences of P. bagrecito and N. peruviana are currently not available. Future molecular analyses of these two species will help resolve relationships among species of Noblella and Psychrophrynella. Until DNA sequences of these two type species become available, new species can be assigned to either genus on the basis of overall morphological similarity. Genetic distances with species whose genes have been sequenced can provide further support for generic assignment.

During a rapid survey of the amphibian fauna of several tributaries of the Inambari River in the Department of Puno, we visited a humid montane forest in the Ollachea Valley. As a result of opportunistic, intensive search of the leaf litter, we found four specimens and recorded the call of a new species of Holoadeninae. Because the advertisement call sounded similar to the calls of P. chirihampatu and P. usurpator (see Catenazzi & Ttito, 2016), we suspected that the new species was a Psychrophrynella. Here we describe this new species, and provide morphological and molecular evidence for its generic allocation.

Methods

We follow Duellman & Lehr (2009) and Lynch & Duellman (1997) for the format of the diagnosis and description, except that the term dentigerous processes of vomers is used instead of vomerine odontophores (Duellman, Lehr & Venegas, 2006). We follow Heinicke et al. (2018) for taxonomic arrangement of genera within subfamilies. We derived meristic traits of similar species from the specimens examined (Appendix 1), from species descriptions, and from published photographs of live or preserved specimens. For codes of collections we used the following acronyms: CORBIDI = Herpetology Collection, Centro de Ornitología y Biodiversidad, Lima, Peru; KU = Natural History Museum, University of Kansas, Lawrence, Kansas, USA; MHNC = Museo de Historia Natural del Cusco, Peru; MUBI = Museo de Biodiversidad del Perú, Cusco, Peru; MHNG = Muséum d’Histoire Naturelle, Genève, Switzerland; and MUSM = Museo de Historia Natural Universidad Nacional Mayor de San Marcos, Lima, Peru.

We fixed and preserved specimens in 70% ethanol. We determined sex and maturity of specimens by observing sexual characters and gonads through dissections. We measured the following variables (Table 1) to the nearest 0.1 mm with digital calipers under a stereomicroscope (see Catenazzi & Ttito, 2016): snout–vent length (SVL), tibia length (TL), foot length (FL, distance from proximal margin of inner metatarsal tubercle to tip of Toe IV), head length (HL, from angle of jaw to tip of snout), head width (HW, at level of angle of jaw), eye diameter (ED), tympanum diameter (TY), interorbital distance (IOD), upper eyelid width (EW), internarial distance (IND), and eye–nostril distance (E–N, straight line distance between anterior corner of orbit and posterior margin of external nares). Fingers and toes are numbered preaxially to postaxially from I–IV and I–V respectively. We determined comparative lengths of toes III and V by adpressing both toes against Toe IV; lengths of fingers I and II were determined by adpressing these fingers against each other. We describe variation in coloration on the basis of field notes and photographs of live frogs. We deposited photographs of live specimens (taken by A. Catenazzi) at the Calphoto online database (http://calphotos.berkeley.edu).

Table 1 Measurements of the type series of Psychrophrynella glauca sp. n.

Characters	Females (n = 2)	Males (n = 1)	
SVL	18.2–19.8	11.3	
TL	8.41–9.5	6.3	
FL	8.2–9.4	5.5	
HL	6.3–6.6	4.3	
HW	5.8–6.3	3.9	
ED	2.0–2.1	1.5	
TY	1.1	0.7	
IOD	2.0–2.1	1.6	
EW	1.3–1.5	1.1	
IND	1.8–2.0	1.0	
E–N	1.5–1.5	1.1	
TL/SVL	0.46–0.48	0.56	
FL/SVL	0.45–0.47	0.49	
HL/SVL	0.33–0.35	0.38	
HW/SVL	0.32	0.35	
HW/HL	0.92–0.95	0.91	
E–N/ED	0.75–0.76	0.73	
EW/IOD	0.65–0.71	0.69	
Note:

Range and average (±standard deviation) measurements (in mm) of males and females of the type series of Psychrophrynella glauca sp. n.

We recorded the advertisement call of an unvouchered male of the new species at the type locality near Thiuni, Distrito Ollachea, Provincia Carabaya, Department of Puno, Peru on August 14, 2017, and recorded air temperature with a quick reading thermometer. No other males of the species were heard calling during our rapid survey. We used a digital recorder (Zoom H2, recording at 48 kHz, 24-bit, WAV format) for field recording, and Raven Pro version 1.4 (Cornell Laboratory of Ornithology, Ithaca, NY, USA) to analyze call variables. We analyzed a single call. We measured the following variables from the oscillogram: note duration and rate, interval between notes, number of pulses, and presence of amplitude modulation. We measured the following variables from the spectrogram: dominant frequency, and presence of frequency modulation or harmonics. Spectral parameters were calculated through fast Fourier transform set at a length of 512 points (Hann window, 50% overlap). Values are reported as averages followed ±standard deviation.

We estimated genetic distances between the new species and other species of Psychrophrynella, as well as species from other genera of Holoadeninae, through analysis of a fragment of the non-coding 16S rRNA mitochondrial gene. We did not conduct phylogenetic analyses because there is uncertainty concerning the taxonomic position of Noblella and Psychrophrynella, and because genetic sequences of their type species (N. peruviana and P. bagrecito) are not available. We used liver tissue from all type specimens (Table 1) to obtain DNA sequences for the new species (Appendix 2 with GenBank accession codes; FASTA Supplemental File). We compared our sequences with those of other species of Psychrophrynella, and with those of Holoadeninae species in related genera (Barycholos, Bryophryne, Holoaden, Microkayla, and Noblella) from GenBank (Appendix 2). We extracted DNA with a commercial extraction kit (IBI Scientific, Peosta, IA, USA). We followed standard protocols for DNA amplification and sequencing (Hedges, Duellman & Heinicke, 2008). We used the 16Sar (forward) primer (5′–3′ sequence: CGCCTGTTTATCAAAAACAT) and the 16Sbr (reverse) primer (5′–3′ sequence: CCGGTCTGAACTCAGATCACGT) (Palumbi et al., 2002). For the polymerase chain reaction (PCR) we used these thermocycling conditions: 1 cycle of 96 °C/3 min; 35 cycles of 95 °C/30 s, 55 °C/45 s, 72 °C/1.5 min; 1 cycle 72 °C/7 min. We used a Veriti thermal cycler (Applied Biosystems, Carlsbad, CA, USA). We purified PCR products with Exosap-IT (Affymetrix, Santa Clara, CA, USA), and shipped purified samples to MCLAB (San Francisco, CA, USA) for sequencing. We aligned sequences using Geneious R8, version 8.1.6 (Biomatters, http://www.geneious.com/) with the MAFFT v7.017 alignment program (Katoh & Standley, 2013), and trimmed sequences to a length of 558 bp. We estimated uncorrected p-distances (i.e., the proportion of nucleotide sites at which any two sequences are different) with the R package “APE” (Paradis, Claude & Strimmer, 2004).

Our research was approved by the Institutional Animal Care and Use Committee of Southern Illinois University Carbondale (protocol #16-006). The Dirección General Forestal y de Fauna Silvestre, Ministerio de Agricultura y Riego issued the permit authorizing this research (permits #0292-2014-MINAGRI-DGFFS/DGEFFS, #029-2016-SERFOR-DGSPFS).

The electronic version of this article in portable document format will represent a published work according to the International Commission on Zoological Nomenclature (ICZN), and hence the new names contained in the electronic version are effectively published under that Code from the electronic edition alone. This published work and the nomenclatural acts it contains have been registered in ZooBank, the online registration system for the ICZN. The ZooBank LSIDs (Life Science Identifiers) can be resolved and the associated information viewed through any standard web browser by appending the LSID to the prefix http://zoobank.org/. The LSID for this publication is: urn:lsid:zoobank.org:pub:B3E69C6D-2669-46A8-A4C5-631C5F1160B2. The online version of this work is archived and available from the following digital repositories: PeerJ, PubMed Central, and CLOCKSS.

Results

Psychrophrynella glauca sp. n. lsid:zoobank.org:act:E815EC45-81B4-46BF-A9A7-3E359DEBDB73 http://zoobank.org/NomenclaturalActs/E815EC45-81B4-46BF-A9A7-3E359DEBDB73.

Holotype

CORBIDI 18729, an adult female from 13.67603 S; 70.46588 W (WGS84), 2,225 m a.s.l., near Thiuni, Distrito Ollachea, Provincia Carabaya, Department of Puno, Peru, collected by A. Catenazzi and A. Ttito on August 14, 2017 (Figs. 1–3; Table 1).

Figure 1 Map of southern Peru indicating the type localities of Peruvian species of Psychrophrynella.

Psychrophrynella bagrecito (black square), P. chirihampatu (black circle), P. glauca sp. n. (white circle), and P. usurpator (triangle).

Figure 2 Photographs of live and preserved specimen of the holotype of Psychrophrynella glauca sp. n.

Live (A, C, E) and preserved (B, D, F) specimen of the holotype, female CORBIDI 18729 (SVL 18.2 mm) in dorsolateral (A, B), dorsal (C, D), and ventral (E, F) views. Photographs by A. Catenazzi.

Figure 3 Palmar and plantar surfaces of the holotype of Psychrophrynella glauca sp. n.

Ventral views of hand (A) and foot (B) of holotype, CORBIDI 18729 (hand length 3.8 mm, foot length 8.2 mm). Photographs by A. Catenazzi.

Paratypes

Three total: one adult male, CORBIDI 18730, one adult female, MUBI 16322, and one juvenile, MUBI 16323 collected at the type locality by A. Catenazzi and A. Ttito on August 14, 2017 (Fig. 4).

Figure 4 Dorsolateral and ventral views of four paratypes of Psychrophrynella glauca sp. n. showing variation in dorsal and ventral coloration.

Female MUBI 16322 (A, B). Male CORBIDI 18730 (E, F). Juvenile MUBI 16323 (G, H). Photographs by A. Catenazzi.

Generic placement

A new species of Psychrophrynella as defined by De la Riva et al. (2017). Frogs of the genus Psychrophrynella are morphologically similar and closely related to Barycholos, Bryophryne, Holoaden, Microkayla, and Noblella (De la Riva et al., 2017; Duellman & Lehr, 2009; Hedges, Duellman & Heinicke, 2008; Heinicke, Duellman & Hedges, 2007; Padial, Grant & Frost, 2014). The new species is assigned to Psychrophrynella rather than to any of the other genera on the basis of overall morphological resemblance with the type species P. bagrecito, including presence of a short fold-like tubercle on the inner edge of the tarsus, call composed of multiple notes, and similarity of molecular data (Table 2). The species with the lowest genetic distance is P. usurpator (16S rRNA uncorrected p-distance: 12.3–12.5%), followed by P. chirihampatu (12.5–12.7%). Species from other genera have genetic distances above 16.7%.

Table 2 Uncorrected p-distance for 16S rRNA between Psychrophrynella glauca sp. n. and related taxa in the subfamily Holadeninae.

	Barycholos pulcher	Bryophryne bakersfield	Bryophryne bustamantei	Bryophryne cophites	Bryophryne phuyuhampatu	Bryophryne quellokunka	Bryophryne tocra	Holoaden luederwaldti	Microkayla boettgeri	Microkayla chaupi	Microkayla chilina	Microkayla guillei	Microkayla wettseteini	Noblella heyeri	Noblella lochites	Noblella myrmecoides	Noblella sp. (SanMartin)	Psychrophrynella chirihampatu	Psychrophrynella glauca MUBI16322	Psychrophrynella glauca (holotype)	Psychrophrynella glauca MUBI16323	Psychrophrynella glauca CORBIDI 18730	Psychrophrynella usurpator	Strabomantis sulcatus	
Barycholos pulcher																									
Bryophryne bakersfield	25.3																								
Bryophryne bustamantei	25.1	5.4																							
Bryophryne cophites	25.5	7.2	10.1																						
Bryophryne phuyuhampatu	25.9	7.5	9.6	7.1																					
Bryophryne quellokunka	26.4	5.4	7.6	6.3	6.0																				
Bryophryne tocra	27.1	7.2	10.0	10.1	10.9	8.3																			
Holoaden luederwaldti	26.6	20.6	21.0	22.0	21.4	21.7	20.6																		
Microkayla boettgeri	24.2	19.9	20.0	21.7	20.3	20.7	20.9	21.8																	
Microkayla chaupi	25.0	18.8	19.6	21.4	20.1	20.0	19.9	22.1	4.7																
Microkayla chilina	25.3	19.7	20.3	21.4	20.1	20.7	20.4	22.0	2.6	4.5															
Microkayla guillei	25.0	19.1	20.1	22.5	21.3	21.3	19.6	20.8	9.3	9.7	9.5														
Microkayla wettseteini	27.8	21.4	22.6	21.3	21.0	20.7	20.8	20.8	14.8	14.5	14.5	12.7													
Noblella heyeri	14.5	14.4	13.9	14.8	13.7	15.4	16.2	18.2	13.9	14.4	13.9	13.9	15.0												
Noblella lochites	23.5	22.6	22.9	23.9	22.4	24.4	24.1	23.1	23.3	23.6	23.3	23.0	25.0	8.9											
Noblella myrmecoides	11.8	13.9	12.3	14.4	12.7	14.9	16.9	15.6	11.2	11.7	11.2	11.2	12.8	11.8	10.6										
Noblella sp. (SanMartin)	25.8	22.2	23.6	24.2	21.2	22.8	22.5	24.0	25.8	26.3	25.7	25.6	25.4	11.7	19.2	6.7									
Psychrophrynella chirihampatu	24.0	22.3	22.0	21.0	22.6	22.2	22.0	22.1	21.5	21.0	21.0	21.4	21.2	19.3	23.1	17.2	25.3								
Psychrophrynella glauca MUBI16322	23.8	18.4	18.3	20.8	20.7	20.2	19.9	20.5	19.0	19.0	19.0	19.3	19.2	16.7	23.4	16.8	24.9	12.5							
Psychrophrynella glauca (holotype)	23.8	18.6	18.5	21.0	20.7	20.3	20.1	20.6	19.1	19.2	19.1	19.4	19.4	16.7	23.3	16.8	24.8	12.7	0.2						
Psychrophrynella glauca MUBI16323	23.8	18.4	18.3	20.8	20.7	20.2	19.9	20.5	19.0	19.0	19.0	19.3	19.2	16.7	23.4	16.8	24.9	12.5	0.0	0.2					
Psychrophrynella glauca CORBIDI 18730	23.8	18.4	18.3	20.8	20.7	20.2	19.9	20.5	19.0	19.0	19.0	19.3	19.2	16.7	23.4	16.8	24.9	12.5	0.0	0.2	0.0				
Psychrophrynella usurpator	24.0	21.6	21.1	22.8	23.0	22.6	21.9	21.3	21.7	21.5	21.3	20.8	20.4	18.9	22.6	16.8	26.3	7.9	12.3	12.5	12.3	12.3			
Strabomantis sulcatus	24.3	19.3	20.8	17.6	17.5	19.3	20.4	20.1	21.4	21.4	20.8	21.9	21.7	17.0	24.2	13.8	24.8	19.8	18.4	18.6	18.4	18.4	20.7		
Note:

Percent genetic distances estimated from the non-coding 16S rRNA mitochondrial fragment (highlighted in gray the genetically most similar species).

Characterization

A species of Psychrophrynella characterized by (1) skin on dorsum smooth to finely shagreen; skin on venter smooth, discoidal fold present; (2) tympanic membrane not differentiated, anteroventral part of tympanic annulus visible below skin; (3) snout very short, bluntly rounded in dorsal view and in profile; (4) upper eyelid lacking tubercles, narrower than IOD; cranial crests absent; (5) dentigerous processes of vomers absent; (6) vocal slits present; nuptial pads absent; (7) Finger I slightly shorter than Finger II; tips of digits bulbous, not expanded laterally; (8) fingers lacking lateral fringes; (9) ulnar tubercles absent; (10) heel lacking tubercles; inner edge of tarsus bearing a short, obliquous fold-like tubercle; (11) inner metatarsal tubercle elliptical, of similar relief and length of prominent, ovoid, outer metatarsal tubercle; supernumerary plantar tubercles absent; (12) toes lacking lateral fringes; webbing absent; Toe V slightly shorter, or about the same length as Toe III; tips of digits not expanded, weakly pointed; (13) dorsum reddish-brown to tan, with dark brown markings, with or without an orange middorsal line extending from tip of snout to cloaca and to posterior surface of thighs; interorbital bar present; flanks brown with dark markings or entirely dark; chest dark brown with bluish-gray flecks; throat and palmar and plantar surfaces grayish-brown with small, bluish-gray flecks; belly and legs red or reddish-brown with bluish-gray flecks; (14) SVL of males 11.3 mm (based on one specimen), SVL of females 18.2–19.8 mm (based on two specimens).

Diagnosis

The new species differs from the three known species of Psychrophrynella by its unique combination of red coloration on ventral surfaces of legs and belly, and profusion of bluish-gray flecks on ventral surfaces of head, body, and legs. Morphologically, it is most similar to P. bagrecito in having a short fold-like tubercle on the inner edge of tarsus, a prominent ovoid outer metatarsal tubercle, discoidal fold present, an elliptical pupil, small size reaching ∼19 mm, and dark brown flanks in at least some specimens. It can be distinguished from P. bagrecito (characters in parenthesis in P. bagrecito) by having smooth skin on venter (areolate), dorsal coloration with broad markings (longitudinal stripes), snout short and bluntly rounded (snout moderately long, rounded in dorsal view and in profile), and ventral coloration in preservative brown with light gray flecks (white to cream with brown mottling). The new species can be distinguished from P. chirihampatu by having reddish-brown to dark brown coloration and bluish-gray flecks on ventral parts (ventral coloration yellow with reddish-brown or gray flecks), Finger I slightly shorter or the same length as Finger II (Finger I shorter than Finger II), inner metatarsal tubercle the same length of outer metatarsal tubercle (inner metatarsal tubercle at least three times the size of outer metatarsal tubercle), more bluntly rounded head (slender and longer head), smaller size reaching 19.8 mm in females (27.7 mm), and advertisement call having 26 notes and a fundamental frequency of 3,027 Hz (up to 68 notes, 2,712 Hz). The new species differs from P. usurpator by its reddish-brown ventral coloration (dull brown, gray or black with cream flecks), smaller SVL reaching 19.8 mm in females (SVL up to 30.5 mm), and by the fold-like tubercle on the inner edge of tarsus being short (long and prominent tubercle).

Description of holotype

Adult female (18.2 mm SVL); head narrower than body, its length 34.6% of SVL; head slightly longer than wide, HL 108.6% of HW; HW 31.9% of SVL; snout very short, bluntly rounded in dorsal and lateral views, ED 31.7% of HL, its diameter 1.3 times as large as its distance from the nostril; nostrils not protuberant, close to snout, directed laterally; canthus rostralis slightly concave in dorsal view, convex in profile; loreal region flat; lips rounded; upper eyelids lacking tubercles; EW 65.0% of IOD; interorbital region flat, lacking cranial crests; E–N distance 75.0% of ED; supratympanic fold weak; tympanic membrane not differentiated, anteroventral part of tympanic annulus visible below skin; postrictal tubercles absent. Choanae round, very small, positioned far anterior and laterally, widely separated from each other; dentigerous processes of vomers and vomerine teeth absent; tongue large, ovoid, not notched.

Skin on dorsum smooth to finely shagreen; dorsolateral folds present only anteriorly and barely visible; skin on flanks and venter smooth; no pectoral or discoidal fold; cloaca not protuberant, cloacal region without tubercles. Ulnar tubercles and folds absent; palmar tubercle flat and oval, approximately the same length but twice the width of elongate, thenar tubercle; supernumerary palmar tubercles absent; subarticular tubercles prominent, ovoid in ventral view, rounded in lateral view, largest at base of fingers; fingers lacking lateral fringes, not webbed; relative lengths of fingers 3 > 4 > 2 ≥ 1 (Fig. 3); tips of digits bulbous, not expanded laterally; forearm lacking tubercles.

Hindlimbs moderately long, TL 46.2% of SVL; FL 45.1% of SVL; upper and posterior surfaces of hindlimbs smooth; heel without tubercles; inner edge of tarsus bearing a short, oblique fold-like tubercle, outer edge of tarsus lacking tubercles; inner metatarsal tubercle elliptical, of similar relief and length of prominent, ovoid, outer metatarsal tubercle; minute plantar supernumerary tubercles weakly defined; subarticular tubercles rounded, ovoid in ventral view; toes lacking lateral fringes, not webbed; toe tips weakly pointed, not expanded laterally; relative lengths of toes 4 > 3 > 5 > 2 > 1 (Fig. 3); FL 45.1% of SVL.

Measurements of holotype (in mm): SVL 18.2, TL 8.4, FL 8.2, HL 6.3, HW 5.8, ED 2.0, TY 1.1, IOD 2.0, EW 1.3, IND 1.8, E–N 1.5.

Coloration of holotype in alcohol: dorsal surfaces of head, body, and limbs grayish tan, with a dark brown X-shaped middorsal mark bordered laterally by a cream line. The interorbital bar is a narrow dark stripe and is bordered anteriorly by a poorly defined cream stripe. There is a dark brown stripe extending from the tip of the snout to above the tympanum and the insertion of forelimb; furthermore, there are two longitudinal dark markings along the line separating the dorsum from the flanks, and dark markings on each side of dorsum near the point of hind limb insertion. The iris is dark gray. The throat is brown anteriorly, fading into pale brown with light gray flecks posteriorly. Chest and belly brown with light gray mottling and large flecks. Ventral parts of limbs reddish-brown with cream mottling and flecks on brachium and thighs, and tan with light gray flecks on antebrachium, crus, and pes. The dorsal surfaces of hind limbs have transverse dark bars. The posterior surfaces of thighs are reddish-brown with a large, dark tan marking surrounding the cloaca and reaching one third the length of thigh, bordered anteriorly by a narrow, cream stripe; the plantar and palmar surfaces are tan, fading into light gray along fingers and toes.

Coloration of holotype in life: similar to coloration in alcohol, but the dorsal coloration varies from beige to brown, and the thighs are reddish-brown with brown mottling. Ventrally, flecks are bluish-gray, largest and most noticeable on chest, and the belly and ventral surfaces of limbs are red or reddish-brown. The iris is dark tan with golden flecks, forming a ring around the pupil.

Variation

Coloration in life is based on field notes and photographs taken by A. Catenazzi of the three paratypes (Fig. 4). These three paratypes have two subocular dark brown spots, which are not visible in the holotype. Furthermore, all three have more extensive dark coloration on flanks, either forming a nearly continuous dorsolateral line connected to the supratympanic marking as in MUBI 16322, or having several dark markings as in CORBIDI 18730, or the entire flank dark as in juvenile MUBI 16323. The latter specimen also has a much darker dorsum than the other type specimens, as well as an orange middorsal line extending from the tip of snout to the cloaca and to the posterior surface of thighs. The ventral coloration of thighs varies from reddish-brown with brown mottling in the two females (the holotype and MUBI 16322), to bright red with little mottling in the male, and orange with little brown mottling in the juvenile.

Advertisement call

A single call of an unvouchered specimen was recorded at 19h45 on August 14, 2017 (Fig. 5). At a Tair = 13.7 °C, the advertisement calls lasted 2,188 ms, and consisted of 26 single-pulsed notes, produced at a rate of 11.88 notes/s. Low amplitude and poor recording quality prevented analysis of the first five notes. In the remaining 21 notes, peak frequency averaged 3,027 ± 22 Hz (range 2,756–3,100 Hz) and increased during calls (F1,19 = 21.3, p < 0.001); peak frequency averaged 2,900 ± 29 Hz for the sixth to 11th note, and 3,078 ± 16 Hz for the last three notes. Amplitude also increased during the call (F1,19 = 6.7, p < 0.017), reaching peak amplitude for notes that had the highest frequency and longest duration (notes 19–21). Average note duration was 15.6 ± 2.4 ms (range 9–53 ms), and the 19th, 20th, and 21st notes had longer duration (38.3 ± 7.9 ms) than all other notes (11.9 ± 1.0 ms).

Figure 5 Advertisement call of Psychrophrynella glauca sp. n.

Advertisement call of an unvouchered male, recorded at the type locality on August 14, 2017 (Tair = 13.7 °C).

Etymology

The specific name glauca is the feminine form of the Latin adjective glaucus, from the ancient Greek noun glaûkos, meaning “bluish-gray,” in reference to the bluish-gray flecks on the ventral parts of body and limbs.

Distribution, natural history, and threats

The four specimens were found in the leaf litter along a descending ridge separating two creeks in the humid montane forest along the road from Thiuni to Ollachea. Sympatric species detected during our quick survey included Gastrotheca testudinea, Pristimantis platydactylus, and an unnamed Pristimantis sp. Much of the original forest vegetation has been replaced by cultivated fields and pasture along the road, but this remnant forest extended from nearly the side of the road to the upper ridge of the mountain. Further advance of agriculture, or clearing of the forest might threaten this species if its distribution is restricted to the Ollachea Valley. In absence of more detailed data regarding its extent of occurrence, and according to the IUCN Red List criteria and categories (IUCN, 2013), we suggest this species to be in the “Data Deficient” category of the Red List.

Discussion

The diversity of small, terrestrial-breeding frogs in the humid grasslands and montane forests of the Tropical Andes has until recently been grossly underestimated (De la Riva et al., 2017). A similar pattern has occurred in the Atlantic forest of Brazil, where the diversity and micro-endemism of the minute terrestrial-breeding Brachychephalus was long unappreciated (Pie et al., 2017; Pie & Ribeiro, 2015; Ribeiro et al., 2017). In Peru, most species of Holoadeninae have been described since 2008 (Catenazzi, Uscapi & von May, 2015; Catenazzi & Ttito, 2016; Catenazzi et al., 2017b; De la Riva, Chaparro & Padial, 2008; De la Riva et al., 2017; Lehr & Catenazzi, 2008; Lehr & Catenazzi, 2009a, 2009b, 2010). Additional, unnamed species of Psychrophrynella have already been identified (Catenazzi, Lehr & von May, 2013; von May et al., 2017), and museum material indicates that several more species might exist among misidentified specimens, such as in the type series of P. bagrecito (De la Riva, Chaparro & Padial, 2008; Duellman & Lehr, 2009). Therefore, we can expect that additional field work, specimen comparisons, bioacoustics and genetic or genomic analyses will reveal many more species of Psychrophrynella and related Holoadeninae genera from the Tropical Andes.

De la Riva et al. (2017) recently allocated all Bolivian species previously assigned to Psychrophrynella, and the Peruvian species P. boettgeri from the Department of Puno, to the new genus Microkayla. Accordingly, the genus Psychrophrynella was left with only three species, P. bagrecito, P. chirihampatu, and P. usurpator, all distributed around the Vilcanota massif and its associated cordilleras in the Peruvian Department of Cusco. Using phylogenomic approaches, Heinicke et al. (2018) validated the allocation of Barycholos, Bryophryne, Euparkerella, Holoaden, Microkayla, Noblella, and Psychrophrynella (but not of Niceforonia, Lynchius, Oreobates, and Phrynopus) within Holoadeninae; we follow their proposed taxonomic arrangement here. With the description of P. glauca, the geographic distribution of Psychrophrynella is extended to the Cordillera de Carabaya in Department of Puno. The Cordillera de Carabaya also contains the type localities of N. peruviana, the type species of Noblella, at Santo Domingo in the upper reaches of a small tributary of the Inambari River, of M. boettgeri at Phara in Province Sandia, and of Bryophryne tocra and Bryophryne wilakunka in Province Carabaya. The Cordillera de Carabaya is thus unique in being home to four of the seven genera of Holoadeninae: Bryophryne, Microkayla, Noblella, and Psychrophrynella. Only three genera are known to occur in the northern Cordillera de Vilcanota and associated cordilleras (Bryophryne, Noblella, and Psychrophrynella), and only two genera in the northern Cordillera de Urubamba (Bryophryne and Noblella), and in the southern Cordillera de Apolobamba (Microkayla and Noblella). Therefore, the Cordillera de Carabaya appears to host substantial beta diversity of Holadeninae, suggesting an intriguing evolutionary history for this group in southern Peru.

Our generic allocation remains tentative in light of an unresolved taxonomic situation regarding Noblella and Psychrophrynella, as previously reviewed (Catenazzi & Ttito, 2016; De la Riva, Chaparro & Padial, 2008; De la Riva et al., 2017). In short, the type species of both genera, N. peruviana and P. bagrecito, respectively, have not been included in phylogenetic analyses due to lack of DNA sequences, and they share several morphological traits, indicating that they might form part of the same clade. Here we have assigned the new species to Psychrophrynella on the basis of general body shape and appearance, overall similarity with the type species P. bagrecito, and similarities in advertisement call with P. chirihampatu and P. usurpator. Furthermore, these two species have the lowest uncorrected p-distances of 16S rRNA in our analysis. As a priority, future work should sample tissues and record advertisement calls of P. bagrecito and N. peruviana, so that multiples approaches can be pursued to determine the phylogenetic relationships of species of Noblella and Psychrophrynella.

Anuran communities in the humid montane forests of southern Peru have undergone sharp reductions in species richness and abundance following epizootics of chytridiomycosis (Catenazzi et al., 2011; Catenazzi, Lehr & Vredenburg, 2014). The disease/host dynamics now seem to be enzootic (Catenazzi et al., 2017a), and although experimental infection trials have shown that terrestrial-breeding frogs can be highly susceptible to chytridiomycosis (Catenazzi et al., 2017a), populations of Strabomantidae generally have not declined as sharply as those of aquatic-breeding sympatric frogs. Thus, chytridiomycosis might not directly threaten P. glauca. A more immediate threat to P. glauca is embodied by hydroelectric projects that are planned or under construction along the San Gabán River. The new dams might directly flood montane forest, or intercept water from streams and rivers that drain the forest, thus reducing habitat quality. These projects are part of many planned dams in the Inambari watershed that threaten to alter fish migrations, biodiversity and geochemical cycles locally and downstream throughout the Amazon basin (Forsberg et al., 2017; Latrubesse et al., 2017). These consequences might not have been properly taken into consideration during the decision-making process evaluating financial interests and the findings of the Environmental Impact Assessment (Rode et al., 2015). We hope that the timely description of new species such as P. glauca will contribute to the conservation of these humid montane forests, and promote mitigating solutions including restoration of degraded forest habitat.

Conclusion

We describe a new species of terrestrial-breeding frog of the family Strabomantidae, and provide evidence for its allocation within the genus Psychrophrynella. The new species P. glauca is only known from its type locality, similarly to most other small Holoadeninae known to occur at high elevations in the Andes of southern Peru and Bolivia. With our description we contribute to a better knowledge of the diversity of this group, and reveal the presence of four genera of Holoadeninae in the Cordillera de Carabaya of southern Peru, suggesting that phylogeographic studies of the Holoadeninae species of this mountain range may shed insights into radiation in this group.

Supplemental Information

Supplemental Information 1 Specimens examined.

Click here for additional data file.

Supplemental Information 2 Gene sequences for molecular analyses.

Genbank accession numbers for the taxa and genes sampled in this study.

Click here for additional data file.

Supplemental Information 3 16S rRNA sequences (FASTA format) for the four types of Psychrophrynella glauca sp. n.

Click here for additional data file.

We thank S. Castroviejo-Fisher, I. De la Riva, and E. Piedra for manuscript revision. We thank A. Blount, S. Cameron, and A. Shepack for lab assistance.

Additional Information and Declarations

Competing Interests

Author Contributions

Animal Ethics

Field Study Permissions

Data Availability

New Species Registration

The authors declare that they have no competing interests.

Alessandro Catenazzi conceived and designed the experiments, performed the experiments, analyzed the data, contributed reagents/materials/analysis tools, prepared figures and/or tables, authored or reviewed drafts of the paper, approved the final draft, field work.

Alex Ttito conceived and designed the experiments, performed the experiments, contributed reagents/materials/analysis tools, authored or reviewed drafts of the paper, approved the final draft, field work.

The following information was supplied relating to ethical approvals (i.e., approving body and any reference numbers):

The Institutional Animal Care and Use Committee at Southern Illinois University Carbondale approved this research.

The following information was supplied relating to field study approvals (i.e., approving body and any reference numbers):

The Dirección General Forestal y de Fauna Silvestre, Ministerio de Agricultura y Riego issued the permit authorizing this research.

The following information was supplied regarding data availability:

The sequences described here are accessible via GenBank accession numbers MG837565 to MG837568 (four sequences). The sequences are also uploaded in Fasta format as Supplemental Dataset Files. The recording of the advertisement call has been deposited at Fonoteca Zoológica (FZ SOUND CODE 11169) and is available at http://www.fonozoo.com/fnz_detalles_registro.php?tipo_registro=2&id=22414.

The following information was supplied regarding the registration of a newly described species:

Publication LSID: urn:lsid:zoobank.org:pub:B3E69C6D-2669-46A8-A4C5-631C5F1160B2;

Species name: urn:lsid:zoobank.org:act:E815EC45-81B4-46BF-A9A7-3E359DEBDB73.

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
