# Peer review of "Psychrophrynella glauca sp. n., a new species of terrestrial-breeding frogs (Amphibia, Anura, Strabomantidae) from the montane forests of the Amazonian Andes of Puno, Peru"

_PeerJ, doi:10.7717/peerj.4444_

## Round 0.1 · original submission · Minor Revisions

· Academic Editor

Minor Revisions

This is a very straight forward species description with strong evidence for the identity of the new taxon. The authors did a good job compiling the data necessary for the species to be named. However, I agree with reviewer number 1, in that genetic distance is not a determinant character for species recognition, but the outcome of a phylogenetic analysis is needed. Given that there is uncertainty in the phylogenetic position of Psychrophrynella, Microkayla and Noblella I strongly recommend to follow advice from reviewer and do a simple phylogenetic analyses with all sequences available for this study and in data deposition places such as GenBank and include a figure in the manuscript.

On the other hand, please attend comments made on the manuscript itself by the three reviewers.

·

Basic reporting

The manuscript has good flow, professional English and is clear. But, the Figures (especially 1 and 3) must be improved, in Fig. 1 I suggest that the legend of species be included within of the Figure and reduce the size of the elevation legend. In Figure 3 is need improve the quality of photographs, the edge of fingers and toes are poor quality and between Toe II and Toe III is possible observe noise it looks like webbing.

Experimental design

As mentioned in the manuscript (attached) for me the major problem is the generic assignment, I suggest that the authors perform a phylogenetic analysis with the genetic data that they obtained and with those available in GenBank for Psychrophrynella and Microkayla, especially for demonstrate that the new species is not Microkayla

Validity of the findings

Linked to the generic assignment the finding of the authors is weakened due that is unclear if the new species is Psychrophrynella or Microkayla.

Comments for the author

Is clear that the author have a new species in hands and that it need be described, but the manuscript should be improved in several aspects before of be published.

·

Basic reporting

The English usage is fairly good.

The structure, literature cited, methods, results and discussion are all properly arranged, comprehensive, appropriate, and clear. Figures are relevant, necessary, and properly labelled.

Experimental design

Being a basically descriptive work, no experimental design was required. Methodology is appropriate and pertinent standards for its different sections (morphometry, genetics, bioacoustics) have been followed.

Validity of the findings

Besides the intrinsecal interest in the finding of a new species, the extension range of the genus Psychrophrynella to the Cordillera de Carabaya points to this region as a hotspot for Holoadeninae diversity and radiation, which represents one of the most interesting contributions of this paper.

Comments for the author

The current manuscript presents an uncomplicated description of a new species of frog in the diverse clade Holoadeninae, which is showing an astonishing increase in species descriptions in the last years. The English usage is fairly good, although I (a non native English speaker either) have made some corrections and suggestions in the MS itself, to improve it. This, together with some comments, I recommend to address in order to fix some unclear things and improve the MS's general comprehension.

Regarding the References section, I wonder whether citation of Pie et al. 2017 is accepted by PeerJ as a valid reference, considering that it is not a publication but a non peer-reviewed piece of text posted at a website.

·

Basic reporting

This is a very good species description. I have only found a few minor issues that are marked in detail in the attached pdf file.

Experimental design

This is not an experiment driven research, but a descriptive and comparative study. The research is rigorous, sound, and following required ethical and legal aspects.

Validity of the findings

The authors present good quality and unambiguous evidence that they have discovered a new species. Conclusions fit well with data and analyses. The only exception is the advertisement call. The authors state the recorded a single call from an unvouchered specimen but they fail to explain how they assigned the recorded sound to the species. Furthermore, at times, the authors talk as if they have recorded more than one call.

Comments for the author

Congratulations. Very good manuscript. My comments are mainly cosmetic.

---

## Round 0.2 · Minor Revisions

· Academic Editor

Minor Revisions

I have read the revised version of Psychrophrynella glauca sp. n., a new species of terrestrial-breeding frogs (Amphibia, Anura, Strabomantidae) from the montane forests of the Amazonian Andes of Puno, Peru and have to note that authors basically refused to do the comments made by one of the reviewers, the only one who suggested changes really.

I am willing to accept the author´s explanation for not performing a phylogeny based on the argument that: “there are no sequences for the type species of Noblella and Psychrophrynella. A phylogeny that does not resolve this long-standing issue would be unnecessary and could create more confusion” However in the abstract they state: “The new species is assigned to Psychrophrynella on the basis of morphological similarity, including presence of a tubercle on the inner edge of the tarsus, call composed of multiple notes, and genetic data (16S rRNA partial sequences)”. You CAN NOT assign a new species to any given genus based on DNA sequences divergences unless you perform a phylogenetic analyses. So, if the authors don´t want to perform an analysis, it´s fine, but then they need to clean the text and remove phrases like this.

With regards to the figures, the reply: “We appreciate the comments regarding Figure 1, but this map has been used in multiple publications without reviewers raising the comments. We consider the suggested improvements to represent personal preferences of the reviewer. We no longer have access to the license of ArcView used to produce the map, making the suggested changes would place the burden of redoing the map in a different software” I find rude and not having a current license as an explanation to not perform changes I must say is not a valid scientific explanation. I believe every publication is worthy of its own figures and I would suggest authors to improve the figures as requested by the reviewer.

·

Basic reporting

As previously I noted the manuscript has good flow, professional English and is clear. Nevertheless, I suggested review the figures 1 and 3. The authors rejected review the figure 1 arguing that "the suggested improvements to represent personal preferences of the reviewer", maybe it is true however also indicates that the authors don`t like improve the manuscript and they prefer use a template as figure 1 that is identical to the used in Catenazzi et al. 2017. ZooKeys 685: 65-81 except in the symbols of the species. Also, the authors say "We have improved the quality of Figure 3" but only erased the noise between Toe II and Toe III, really they did not take new photos.

Experimental design

As mentioned previously for me the major problem is the generic assignment, I suggested that the authors performed a phylogenetic analysis with the genetic data that they obtained and with those available in GenBank for Psychrophrynella and Microkayla, especially for demonstrate that the new species is not Microkayla. I am not expert in these clades but consider that the phylogenetic tree improve the manuscript, I agree with the authors in that they have a new species of Psychrophrynella but is really important that they perform the phylogenetic analysis. If the authors reject perform the tree they should eliminate the genetic distances as generic assignment because the genetic distances are only adequate to match specimens to groups, not for species delimitation, and even less for trying to assign specimens to clades. Phylogeny and divergence is not the same. You can be very divergent from your sister group, and yet be the sister group, or can be very similar to a very distant relative.

Validity of the findings

Linked to the generic assignment the authors could improve the manuscript if perform a phylogenetic tree. Also, add a phylogenetic tree establish a precedent in the knowledge of these genera and maybe for the widespread of phylogenetic trees in future taxonomic works for these genera.

Comments for the author

I consider that the author have a new species in hands but is important that the authors improve the manuscript, all the comments in review were for improve the manuscript and never negatively but the authors really not make a significant effort for add the comments and suggestions.

---

## Round 0.3 · accepted · Accept

· Academic Editor

Accept

The author´s have made the suggested changes on this manuscript.